# Fatty Acid-Binding Proteins Aggravate Cerebral Ischemia-Reperfusion Injury in Mice

**DOI:** 10.3390/biomedicines9050529

**Published:** 2021-05-10

**Authors:** Qingyun Guo, Ichiro Kawahata, Tomohide Degawa, Yuri Ikeda-Matsuo, Meiling Sun, Feng Han, Kohji Fukunaga

**Affiliations:** 1Department of Pharmacology, Graduate School of Pharmaceutical Sciences, Tohoku University, 6-3 Aramaki-Aoba, Aoba-Ku, Sendai 980-8578, Japan; guo.qingyun.r2@dc.tohoku.ac.jp (Q.G.); kawahata@tohoku.ac.jp (I.K.); tomohide.degawa.r2@dc.tohoku.ac.jp (T.D.); reiko.fukunaga.e2@tohoku.ac.jp (M.S.); 2Laboratory of Pharmacology, Department of Clinical Pharmacy, Faculty of Pharmaceutical Sciences, Hokuriku University, Kanagawa-Machi, Kanazawa 920-1181, Japan; y-matsuo@hokuriku-u.ac.jp; 3School of Pharmacy, Nanjing Medical School, Nanjing 211166, China; fenghan169@njmu.edu.cn

**Keywords:** ischemia, FABP3, FABP5, FABP7, mPGES-1, PGE_2_

## Abstract

Fatty acid-binding proteins (FABPs) regulate the intracellular dynamics of fatty acids, mediate lipid metabolism and participate in signaling processes. However, the therapeutic efficacy of targeting FABPs as novel therapeutic targets for cerebral ischemia is not well established. Previously, we synthesized a novel FABP inhibitor, i.e., FABP ligand 6 [4-(2-(5-(2-chlorophenyl)-1-(4-isopropylphenyl)-1H-pyrazol-3-yl)-4-fluorophenoxy)butanoic acid] (referred to here as MF6). In this study, we analyzed the ability of MF6 to ameliorate transient middle cerebral artery occlusion (tMCAO) and reperfusion-induced injury in mice. A single MF6 administration (3.0 mg/kg, per os) at 0.5 h post-reperfusion effectively reduced brain infarct volumes and neurological deficits. The protein-expression levels of FABP3, FABP5 and FABP7 in the brain gradually increased after tMCAO. Importantly, MF6 significantly suppressed infarct volumes and the elevation of FABP-expression levels at 12 h post-reperfusion. MF6 also inhibited the promotor activity of FABP5 in human neuroblastoma cells (SH-SY5Y). These data suggest that FABPs elevated infarct volumes after ischemic stroke and that inhibiting FABPs ameliorated the ischemic injury. Moreover, MF6 suppressed the inflammation-associated prostaglandin E_2_ levels through microsomal prostaglandin E synthase-1 expression in the ischemic hemispheres. Taken together, the results imply that the FABP inhibitor MF6 can potentially serve as a neuroprotective therapeutic for ischemic stroke.

## 1. Introduction

Ischemic stroke is the most common cerebrovascular disease, accounting for approximately 80% of all strokes and causing a large number of deaths and disabilities worldwide [1,2]. Ischemic stroke is caused by the blockage of the artery supplying blood to the brain, and an insufficient blood supply triggers a series of complex neurochemical responses including excitotoxicity, oxidative stress and inflammation, which ultimately leads to the death and dysfunction of brain cells [3,4,5]. Even though the administration of intravenous thrombolysis with recombinant tissue plasminogen activator is the only clinically approved drug therapy that can effectively treat ischemic stroke, a short therapeutic window and intracerebral hemorrhagic complications limit its availability [5]. Therefore, alternative neuroprotective drugs are still required for treating acute-phase cerebral ischemia. 

Fatty acid-binding proteins (FABPs) are expressed in various tissues in a highly specific manner, where they regulate fatty acid uptake, transport and metabolism [6], and play vital roles in the pathogenesis of many common diseases [7]. Most mammals produce 12 distinct subtypes of FABPs, although humans produce up to 10 [8], three of which are expressed in the brain, including FABP3 (heart-type), FABP5 (epidermal-type) and FABP7 (brain-type) [9,10]. These three FABPs are expressed at specific stages during brain development. For example, FABP3 is not expressed during embryonic period and its expression gradually increases in rodent brains after birth, whereas FABP5 and FABP7 show an opposite pattern and decrease postnatally in rodents [9]. FABP homeostasis is critical for normal brain development and functions at different stages, and FABP imbalances can cause various neurodegenerative and neuropsychiatric disorders. FABP5 and FABP7 mRNA levels were higher in the cortexes of postmortem brains from schizophrenic patients than in those from healthy controls, and similar results were found for FABP7 mRNA in postmortem brains from patients with autism spectrum disorder [9]. Higher FABP3 levels have been observed in the cerebrospinal fluid and serum of patients with Alzheimer’s disease, dementia with Lewy bodies or Parkinson’s disease [11,12]. FABP3 deficiency prevents nicotine-induced preference behavior [13,14]. The FABP3 deficiency also completely abolished the induction of Parkinson’s syndrome after 1-methyl-4-phenyl-1,2,3,6-tetrahydropiridine (MPTP) treatment in mice [15]. 

Developing specific ligands for each FABP is essential for studying the mechanisms of FABPs in neurological diseases. For example, the FABP4 ligand BMS309403 has been used as a potent and selective biphenyl azole inhibitor to improve ischemia/reperfusion (I/R) injury of the brain [16] and kidneys [17] in mice. Recently, we have developed a series of FABP3 ligands, based on BMS309403 [18]. Among these ligands, we confirmed that MF1 [19] and MF8 [20] could target FABP3 and improve motor deficits and cognitive impairments in a mouse model of MPTP-induced Parkinson’s disease. Therefore, we hypothesized that FABP ligands also have neuroprotective effects to rescue neurons from ischemic injury. 

In this study, we used a newly synthesized ligand 6 [4-(2-(5-(2-chlorophenyl)-1-(4-isopropylphenyl)-1H-pyrazol-3-yl)-4-fluorophenoxy)butanoic acid] (named here as MF6), which has a high affinity for FABP7 (dissociation constant [Kd] value: 20 ± 9 nM) [21] and has a much weaker affinity for FABP3 (Kd value: 1038 ± 155 nM) and FABP5 (Kd value: 874 ± 66 nM) [22,23], to examine its therapeutic effects on ischemic stroke. We defined the elevation of FABP3, FABP5 and FABP7 expression after I/R injury in mice brain and the neuroprotective effects of FABP inhibitor MF6. Our results provided a new target protein for ischemic stroke therapeutics and demonstrated the potential of FABP ligands as neuroprotective therapeutic drugs.

## 2. Materials and Methods

### 2.1. Animals

Male ICR mice (Institute of Cancer Research (ICR) mice, Slc:ICR) (5 weeks old, 25–30 g) were purchased from Japan SLC, Inc. (Shizuoka, Japan). The animals were housed under conditions of constant temperature and humidity, kept on a 12-h light-dark cycle (lights on: 09:00–21:00) and fed ad libitum. All procedures for handling animals complied with the Guide for Care and Use of Laboratory Animals and were approved by the Experimentation Committee of Tohoku University Graduate School of Pharmaceutical Sciences [2019PhLM0-021 (approved date: 1 December 2019) and 2019PhA-024 (approved date: 1 April 2019)]. 

### 2.2. Surgical Procedures Used for Establishing Transmit Middle Cerebral Artery Occlusion (Tmcao) and Reperfusion

Mice were randomly assigned to tMCAO and sham groups. The mouse model of tMCAO was generated as previously described [24]. Briefly, adult male ICR mice were anesthetized via intraperitoneal injection with a combination of 0.3 mg/kg body weight medetomidine (Domitol, Meiji Seika Pharma Co., Ltd., Tokyo, Japan), 4.0 mg/kg midazolam (Dormicum, Astellas Pharma Inc., Tokyo, Japan) and 5.0 mg/kg butorphanol (Vetorphale, Meiji Seika Pharma Co., Ltd.). Following this, tMCAO surgical procedure was performed as follows: a silicone-coated 6–0 suture (602356PK10, Doccol Corporation, Sharon, MA, USA) was inserted from the right external carotid artery to the internal carotid artery, extending to the origin of a middle cerebral artery, for 2 h. After 2 h, the suture was removed, allowing reperfusion to occur. Mice in the sham-operation group underwent the same procedure, except for the suture insertion. Following reperfusion, mice were sacrificed at the indicated time. A homoeothermic heating blanket was used to the maintain core body temperature in each mouse at 37 °C during the I/R operation. Regional cerebral blood flow (rCBF) was monitored by laser-doppler flowmetry (FLOC1, OMEGAWAVE, Tokyo, Japan) to confirm whether the right hemisphere was in an ischemic state. When CBF was reduced by approximately 70–90%, the surgery was considered successful as previously described [25]. 

### 2.3. Drug Treatment

MF6 (FABP ligand 6) was synthesized from BMS309403 [23], and its chemical structure is shown in Figure 1A. MF6 was suspended in 0.5% carboxymethyl cellulose (CMC; Wako, Osaka, Japan) and administered per os (p.o.) at different doses (0.5, 1 or 3 mg/kg) just before the experiments were initiated, according to the experimental schedule described in Figure 1B–E. The corresponding control group was orally administered an equivalent volume of 0.5% CMC.

### 2.4. Infarct Volume Evaluation

After 24 h of reperfusion, the mice were decapitated, and their brains were rapidly removed and cooled to −30 °C for 10 min. The brains were sliced into five sections (2-mm thick), incubated in 1% 2,3,5-triphenyltetrazolium chloride (TTC; Wako, Osaka, Japan) for 20 min at 37 °C and then soaked overnight in 4% paraformaldehyde (PFA; Wako, Osaka, Japan). The infarcted areas appeared white, whereas the non-infarcted regions appeared red. Infarct volumes were measured using ImageJ software and were expressed as a percentage of the total hemisphere [24].

### 2.5. Neurological Score

Neurological function impairment was also evaluated after 24 h of reperfusion using a neurological deficit grading system with a scale ranging from 0 to 4, as described previously [24]. The following scale was used as a rating system: 0, normal motor function; 1, forelimb flexion when lifted by the tail; 2, circling to the contralateral side when held by the tail on a flat surface, but a normal posture at rest; 3, spontaneous leaning towards to the contralateral side when moving freely; 4, no spontaneous motor activity with an apparent reduction in consciousness. The mice in the sham group exhibited no manifestations of neurological deficits.

### 2.6. Western Blot Analysis

Following decapitation, a second coronal slice was dissected at 12 h after reperfusion, and regions of the right hemisphere (the ischemic side, ipsilateral) and left hemisphere (the contralateral side) were selected and stored at −80 °C Frozen samples were homogenized in lysis buffer (50 mM Tris–HCl, pH 7.4, 0.5% Triton X-100, 4 mM EGTA, 10 mM EDTA, 1 mM Na_3_VO_4_, 40 mM Na_2_P_2_O_7_·10H_2_O, 50 mM NaF, 100 nM calyculin A, 50 µg/mL leupeptin, 25 µg/mL pepstatin A, 50 µg/mL trypsin inhibitor and 1 mM dithiothreitol). The samples were then centrifuged at 12,000 rpm for 10 min at 4 °C to remove insoluble material. Protein concentrations were determined using Bradford’s assay, and samples were boiled for 3 min at 100 °C with 6× Laemmli’s sample buffer [26]. 

For electrophoresis, equal amounts of proteins were loaded on 15% sodium dodecyl sulfate-polyacrylamide gels and transferred to a polyvinylidene difluoride membrane for 2 h. After blocking with T-TBS solution (50 mM Tris–HCl, pH 7.5, 150 mM NaCl and 0.1% Tween 20) containing 5% fat-free milk powder for 1 h at room temperature, the membranes were incubated overnight at 4 °C with primary antibodies against the proteins of interest. The following working dilutions were used for the indicated monoclonal antibodies, per manufacturer’s suggestions: mouse anti-FABP3 (1:1000; Hycult Biotech, HM2016, Uden, NLD), goat anti-FABP5 (1:1000; R&D Systems, AF3077, Minneapolis, MN, USA), goat anti-FABP7 (1:1000; R&D Systems, AF3166, Minneapolis, MN, USA), rabbit anti-mPGES-1 (1:200; Cayman Chemical, 160140, Ann Arbor, MI, USA) and mouse anti-β-actin (1:5000; Sigma, A5441, St Louis, MO, USA). After washing, membranes were incubated with appropriate secondary antibodies diluted in T-TBS solution for 2 h, at room temperature. The membranes were developed using an enhanced chemiluminescence immunoblotting detection system (Amersham Biosciences, NJ, USA) and visualized with a Luminescent Image Analyzer (LAS-4000 mini, Fuji Film, Tokyo, Japan). The densities of the bands were analyzed with ImageJ software (NIH, Bethesda, MA, USA).

### 2.7. Immunofluorescence Staining

Mice were anesthetized and transcardially perfused 12 h after reperfusion with ice-cold phosphate-buffered saline (PBS) immediately followed by 4% PFA, as previously described. Mouse brains were removed and fixed in 4% PFA overnight at 4 °C The brain samples were cut into 50 µm coronal sections using a vibratome (Dosaka EM Co. Ltd., Kyoto, Japan). Sections were washed in PBS for 30 min, permeabilized in PBS with 0.1% Triton X-100 for 2 h and blocked in PBS containing 1% BSA and 0.3% Triton X-100 for 1 h at room temperature [26]. The brain samples were then incubated with the following primary antibodies in blocking solution for 3 d at 4 °C: mouse anti-FABP3 (1:500), goat anti-FABP5 (1:500), goat anti-FABP7 (1:500), rabbit anti-mPGES-1 (1:200) and antibodies against cellular markers (1:500). After washing with PBS, the brain sections were incubated with Alexa Flour-conjugated secondary antibodies overnight at 4 °C. After sufficient washing with PBS, the brain sections were mounted on slides with Vectashield (Vector Laboratories, Inc., Burlingame, CA, USA). Immunofluorescent images were analyzed using a confocal laser scanning microscope (Nikon, Tokyo, Japan).

### 2.8. Measuring MF6 Concentrations in the Blood and Brain

MF6 (3 mg/kg, p.o.) was administered to the mice directly (sham group) or 30 min after reperfusion (I/R group). Blood was collected from the tail vein for measurement at specific time intervals after injection. At 1, 4 and 48 h after injection, the mice were anesthetized and perfused with ice-cold PBS to remove blood to obtain brain samples, which were stored at −80 °C until use. The blood in heparinized tubes was centrifuged at 16,500× *g* for 10 min at 4 °C, after which the supernatants were collected as plasma. Plasma samples (10 µL) were deproteinized by adding 250 µL of acetonitrile containing ligand 2 (2 ng/mL) as an internal standard, followed by vortexing and sonication. The samples were centrifuged at 16,500× *g* for 10 min at 4 °C. After centrifugation, the supernatants were evaporated to dryness with a centrifugal concentrator (CC-105, TOMY, Tokyo, Japan), and then the residues were dissolved in 20 µL 80% acetonitrile. For brain sample preparations, 1 mL of acetonitrile was added to 1.5 mL tubes containing brain hemisphere sections, after which the mixtures were homogenized with an ultrasonic homogenizer (SONIFIER Model: 250-Advanced, Branson, CT, USA) for 1 min. Then, the samples were centrifuged at 16,500× *g* for 10 min at 4 °C. Ten micro-liters of each supernatant were processed following the same procedure used for the plasma samples.

Ultra-performance liquid chromatography (UPLC; Ultimate 3000, Dionex) was performed with an ACQUITY UPLC® BEH C18 column (2.150 mm, 1.7 µm, Waters, Milford, MA, USA) maintained at 40 °C, with a flow rate of 400 µL/min (0–1.0 min, 2.0–3.0 min) or 600 µL/min (1.0–2.0 min). Mobile phase A was composed of water containing 0.1% formic acid, and mobile phase B was composed of acetonitrile containing 0.1% formic acid. The following gradient program was used: 0–1.0 min, 80% B; 1.0–2.0 min, 98% B; 2.0–3.0 min, 80% B. An injection volume of 1 µL was used for analysis.

Mass spectrometry (MS) was performed using a TSQ Vantage mass spectrometer (Thermo Fisher Scientific, Waltham, MA, USA) with an electrospray ionization inter-face. Quantitative analysis was performed in selected-reaction monitoring mode (ligand 6; m/z [M + H]^+^ 493.2 > 407.2, ligand 2; m/z [M + H]^+^ 479.1 > 393.1).

### 2.9. Measuring Brain PGE_2_ Concentrations by Performing Enzyme-Linked Immunosorbent Assays (ELISAs)

Prostaglandin E_2_ (PGE_2_) concentrations in the ipsilateral and contralateral hemi-spheres (Figure 4A) were determined using a Prostaglandin E_2_ ELISA Kit (Cayman Chemical, 514010, Ann Arbor, MI, USA). Samples were collected into liquid nitrogen as described above and weighed. Fifty microliters homogenization buffer (0.1M phosphate, pH 7.4, containing 1 mM EDTA and 10 µM indomethacin) was added to 1 mg of each tissue, the samples were homogenized and centrifugated at 10,000× *g* for 15 min at 4 °C, and then PGE_2_ was extracted from the supernatant and quantitated according to the manufacturer’s protocol.

### 2.10. Cell Culture and Luciferase Reporter Assay

Human neuroblastoma cells (SH-SY5Y) were grown in Dulbecco’s modified Eagle’s medium (DMEM, Wako) supplemented with 15% fetal bovine serum (FBS, Gibco, CA, USA) and 1% penicillin-streptomycin at 37 °C in a humidified incubator with 5% CO2/95% air. Human genomic DNA extracted from HEK293 cells was used to amplify the FABP5 promoter fragment (positions -1250/-1) and subcloned into pGL3-Basic-luciferase vector (Promega, Madison, WI, USA). All cloned DNA fragments were confirmed by DNA sequencing. SH-SY5Y cells in 35 mm dishes were transfected with 2 μg of FABP5-pGL3 vector, as well as 50 ng of renilla luciferase plasmid (internal control) for 6 h using lipofectamine LTX and Plus Reagent (Invitrogen, Carlsbad, CA, USA) according to the manufacturer’s protocol. After transfection, cells were treated with BSA-AA (arachidonic acid) and MF6 and maintained in D-MEM (1% penicillin-streptomycin) without FBS for 24 h. AA (Sigma, 10931, St Louis, MO, USA) and BSA (Sigma, A7030, fatty acid free) were used to prepare BSA-AA complexes at a 1:5 ratio mixed in binding buffer (10 mM Tris-HCl (pH 8.0), 150 mM NaCl) at 37 °C for 30 min. Firefly luciferase and renilla luciferase activities were measured with the dual-luciferase reporter assay kit (Promega) using a luminometer (Gene Light 55, Microtec, Funabashi, Japan). Relative luciferase activity was expressed as the ratio of firefly luciferase activity to renilla luciferase activity [25].

### 2.11. Statistical Analysis

The results are presented as box and whisker plots (median, first and third quartile, range), overlaid by dot plot of the raw data. Statistical analysis was conducted using SigmaPlot® version 14 (SYSTAT Software Inc., San Jose, CA, USA). Data from all experiments were analyzed by one-way analysis of variance (ANOVA) followed by Dunnett’s test for multiple comparisons (results of concentration gradient and time gradient), or two-way ANOVA followed by Student–Newman–Keuls test for multiple comparisons (MF6 treatment groups). A value of *p* < 0.05 was considered to reflect statistically significant differences.

## 3. Results

### 3.1. Pharmacokinetics of MF6 in the Blood and Brain after I/R

We initially determined whether MF6 could penetrate the blood–brain barrier (BBB) as a prerequisite step for evaluating the therapeutic effects of MF6 on cerebral ischemic injury. We measured MF6 concentrations in the plasma and brain tissues of mice after administering MF6 (3 mg/kg). The results revealed that MF6 easily entered the bloodstream. MF6 concentrations in the plasma of sham-operated mice increased rapidly within the first 3 h (15 min: 88.88 ± 25.46 nM; 3 h: 514.95 ± 66.12 nM), reached the maximum value after approximately 4 h (Cmax = 522.18 ± 74.33 nM) and then gradually decreased to 47.14 ± 7.50 nM by 48 h (Figure 2A). However, concentrations of MF6 in the brain did not increase in mice with I/R-induced BBB dysfunction (Figure 2B), indicating that MF6 penetrates to brain even in non-ischemic mice. Moreover, no significant difference was found in MF6 concentrations between the contralateral and ipsilateral portions of ischemic brains (Figure 2B).

### 3.2. MF6 Reduced Infarct Volumes and Ameliorated Neurological Deficits in I/R Mice

We then examined whether MF6 treatment could reduce cerebral ischemic injury. Mice were treated with or without MF6 (0.5, 1 or 3 mg/kg) 30 min after reperfusion. Mice with I/R surgery showed large infarct volumes (70.28±3.19%) and MF6 treatment signif-icantly reduced infarct volumes in a concentration-dependent manner (1 mg/kg group: 50.13 ± 5.70%, *p* = 0.021 vs. I/R group; 3 mg/kg group: 43.26 ± 4.63%, *p* < 0.001 vs. I/R group; Figure 3A,B). Moreover, MF6-treated I/R mice showed significantly decreased neurological deficits compared to those of the I/R group treated without MF6 (1mg/kg group: *p* = 0.034; 3 mg/kg group, *p* = 0.014 vs. I/R group; Figure 3C). Next, we tested the optimal timing of MF6 treatment and confirmed that the time of MF6 administration after reperfusion was crucial for the therapeutic effects, with earlier treatment resulting in better outcomes. Administering MF6 30 min after reperfusion was more effective (46.64 ± 3.01%, *p* = 0.007 vs. I/R group, 76.66 ± 3.10%), than administering MF6 1 h or 2 h after reperfusion (60.70±9.88%, *p* = 0.231; 63.98 ± 8.80%, *p* = 0.403, respectively; Figure 3D–F).

To confirm the long-lasting protective effects of MF6 on I/R-induced cerebral infarct after the single administration, we evaluated the brain infract volumes and neurological scores at 7 days after reperfusion. The MF6 administration (3 mg/kg) reduced the cerebral infarction and improved the neurological deficits on day 7 after reperfusion (25.93 ± 2.09%, *p* < 0.001 vs I/R group, 55.79 ± 3.32%; Figure 3H,I) and improved the mortality of mice caused by ischemia (*p* = 0.0276 vs. I/R group; Figure 3J). Furthermore, we confirmed the protective effects of MF6 pre-administration in which MF6 was administered once 30 min before ischemia surgery. The MF6 pre-administration significantly reduced the infarct volume (I/R group: 72.88 ± 6.24%) at 1 mg/kg (47.61 ± 3.00%, *p* = 0.012 vs. I/R group) and 3 mg/kg (39.31 ± 5.11%, *p* < 0.001 vs. I/R group) (Appendix A Appendix A). Taken together, MF6 protects brain against I/R injury. Since the effects of 3 mg/kg MF6 treatment were re-productive, we chose the dose of 3 mg/kg as the optimum concentration for the subsequent experiments.

### 3.3. I/R Induced FABP3, FABP5 and FABP7 Protein Expression in Mouse Brains

As previous findings indicated that the FABP5 and FABP7 proteins were significantly upregulated in post-ischemic adult monkey brains [27,28], we hypothesized that the FABP3, FABP5 and FABP7 proteins might also be upregulated in ischemic mouse brains. We measured protein-expression levels in the half-brain region including the cortex and striatum (Figure 4A), as previously described [25]. In the right ischemic hemisphere (ipsilateral area), FABP3 significantly increased at 12 h after reperfusion (*p* = 0.004 vs. sham group) and maintained high protein-expression levels until 48 h (24 h: *p* = 0.024 vs. sham group; 48 h: *p* = 0.047 vs. sham group). I/R also significantly upregulated FABP3 expression in the left, non-ischemic hemisphere (contralateral area; 12 h: *p* = 0.001 vs. sham group; 24 h: *p* = 0.047 vs. sham group), although the trend was weaker than observed with the ipsilateral areas and declined after 24 h (Figure 4C). Moreover, FABP5 and FABP7 expression were also upregulated in ipsilateral brain tissues (Figure 4D,E), and their expression levels increased faster than that of FABP3 (FABP3: 6 h: *p* = 0.088 vs. sham group) and had significantly increased at 6 h post-reperfusion (FABP5: *p* = 0.016 vs. sham group; FABP7: *p* = 0.048 vs. sham group). Specifically, FABP7 expression did not peak until 24 h. However, the rate of FABP7 upregulation was stronger than that of FABP5 in contralateral brain tissues. All considered, these results show that I/R upregulated the FABP3, FABP5 and FABP7 proteins in a time-dependent manner in the whole brains of mice.

### 3.4. I/R-Induced FABPs Were Expressed in Specific Cells in the Cortex

We investigated the phenotypes of FABP3-, FABP5- and FABP7-positive cells in the cortical area of the penumbra (Figure 5A) at 12 h after reperfusion. We co-stained the FABP3, FABP5 and FABP7 proteins and cell type-specific markers, which included NeuN for neurons, GFAP for astrocytes, Olig2 for oligodendrocytes and Iba1 for microglia. In non-ischemic mouse cortexes, FABP3 was almost co-expressed with NeuN (Figure 5B1) but not with Iba1 (Appendix A), indicating that FABP3 was localized to neurons. I/R-induced FABP3 expression also occurred almost in neurons (Figure 5B2). A previous report showed that FABP5 was expressed both in neurons and glial cells [10]. In agreement, we found that FABP5 was expressed in Olig2-positive cells (Figure 5D1), expressed at a slightly lower level in NeuN-positive cells (Figure 5C1) and weaker still in GFAP-positive cells (Appendix A) in non-ischemic mouse cortexes, but that FABP5 was not co-expressed with Iba1 (Appendix A). After I/R, FABP5 was significantly expressed in oligodendrocytes (Olig2^+^ cells, Figure 5D2) and neurons (NeuN^+^ cells, Figure 5C2), and marginally expressed in astrocytes (GFAP^+^ cells, Appendix A). Furthermore, FABP7 was usually expressed in astrocytes (GFAP^+^ cells, Figure 5E1) and oligodendrocytes (Olig2^+^ cells, Figure 5F1) in the cortex, and I/R-induced FABP7 expression was mainly observed in astrocytes (Figure 5E2) but was not obviously induced in oligodendrocytes (Figure 5F2). In addition, FABP7 was not co-expressed with Iba1 (Appendix A).

### 3.5. MF6 Suppressed FABP3, FABP5 and FABP7 Protein Upregulation in I/R Mouse Brains

FABP3, FABP5 and FABP7 expression levels significantly increased at 12 h post-reperfusion, without particularly serious destruction to the structures of brain tissues. Therefore, we next determined whether MF6 could prevent activation of the FABP3, FABP5 and FABP7 proteins at 12 h after reperfusion. MF6 slightly attenuated the increase in FABP3 protein expression (*p* = 0.034 vs. vehicle-treated I/R group) in the ischemic area (ipsilateral), but not in the contralateral area (*p* = 0.851 vs. vehicle-treated I/R group), as shown in Figure 6B. FABP5 and FABP7 expression in ischemic area were significantly suppressed by MF6 treatment (both: *p* < 0.001 vs. vehicle-treated I/R group), and FABP5 expression in the contralateral area was also significantly suppressed (*p* = 0.015 vs. vehicle-treated I/R group), but FABP7 was not (*p* = 0.051 vs. vehicle-treated I/R group) (Figure 6C,D). Moreover, MF6 did not affect FABP3, FABP5 and FABP7 protein levels in non-ischemic mouse brains (sham mice). The present Western blotting data showed that MF6 ameliorated the effects of I/R injury by inhibiting FABP3, FABP5 and FABP7 upregulation.

### 3.6. MF6 Prevented AA-Induced FABP5 Upregulation in SH-SY5Y Cells

To elicit the mechanism of MF6 on FABP induction, we focused on FABP5 gene expression because the neuronal expression of FABP5 is pronounced compared to FABP3. Moreover, the stimulation with arachidonic acid (AA) increases the activation of PPARβ/δ in MCF-7 cells [29] and PPARβ/δ agonist, GW0742 up-regulates FABP5 expression in PC3M cells [30]. SH-SY5Y human neuroblastoma cells were used to investigate the effects of AA on FABP5 transcriptional activity. When SH-SY5Y cells were exposed to AA for 24 h, the FABP5 promoter activity increased significantly (30 µM AA, 1.89-fold the control cells level, *p* < 0.001) and AA effects was bell shape. (Figure 7A). MF6 treatment with 1 µM inhibited the increased FABP5 promoter activity by AA (*p* = 0.001 vs. vehicle-treated AA group, Figure 7B). Taken together, MF6 reduces FABP5 protein expression by inhibiting FABP5 transcription activity.

### 3.7. MF6 Suppressed the Microsomal Prostaglandin E Synthase-1 (mPGES-1)–PGE_2_ Signaling Pathway in I/R Mouse Brains

Post-ischemic inflammation is an important part of the injury mechanism occurring in ischemic stroke. Accumulation of PGE_2_ and its synthase mPGES-1 aggravates cerebral I/R injury [31]. Therefore, we examined changes of mPGES-1 protein expression and PGE_2_ levels after reperfusion in mice. Our results showed that I/R significantly increased mPGES-1 protein expression in ischemic and contralateral areas, and both peaked at 24 h after reperfusion (*p* < 0.001 vs. sham group) (Figure 8A). After 12 h of reperfusion, the PGE_2_ levels in ischemic brains were significantly higher than those of sham-operated brains (ipsilateral: *p* < 0.001 vs. vehicle-treated sham group, contralateral: *p* = 0.003 vs. vehicle-treated sham group) (Figure 8D). Moreover, MF6 treatment significantly reversed I/R-induced increases in mPGES-1 expression (*p* = 0.001 vs. vehicle-treated I/R group) and PGE_2_ levels (*p* = 0.001 vs. vehicle-treated I/R group) in ipsilateral areas without contralateral area (mPGES-1: *p* = 0.435, PGE_2_: *p* = 0.304; Figure 8C,D). By immunostaining mPGES-1-positive cells with cell-type specific markers, we observed that mPGES-1 was expressed in NeuN^+^ neurons but not in GFAP^+^ astrocytes or Olig2^+^ oligodendrocytes in the cortexes of sham mice (Figure 8B) and that I/R-induced increases also occurred in neurons.

## 4. Discussion

In this study, we demonstrated that MF6 decreased brain infarction volumes and neurological deficits in mice subjected to tMCAO and reperfusion by inhibiting FABP3, FABP5 and FABP7 expression in the brain. The FABP inhibitor MF6 inhibited the mPGES-1-dependent induction. These results supported our hypothesis that brain FABPs are key molecules in ischemic stroke and that their excessive activation exacerbates brain inflammation through mPGES-1. Therefore, our findings show that MF6 was beneficial for treating ischemic stroke.

In this study, we first discovered that I/R significantly induced FABP3, FABP5 and FABP7 protein expression in the mouse brain, especially that of FABP5 and FABP7. Our results were consistent with several previous findings showing that FABP5 and FABP7 proteins were abundantly expressed in the post-ischemic hippocampus in adult monkeys [28] and that FABP7 gradually increased in the cerebellar cortex of adult monkeys after ischemia, reaching a maximum at day 15 [27]. These results imply that FABP3, FABP5 and FABP7 protein induction appears to exhibit beneficial of detrimental effects for ischemic injury. Moreover, co-staining FABPs in different cell types in the cortical area of the penumbra suggests that increased FABP3, FABP5 and FABP7 expression occurred only in specific cells expressing these proteins before ischemia. FABP3 immunoreactivity increased in neurons, whereas FABP7 was predominantly expressed and increased in astrocytes. In contrast, FABP5 expression increased both in neurons and oligodendrocytes. The localization of FABP3, FABP5 and FABP7 are consistent with previous studies [10,11,32,33].

Under pathological conditions, elevated FABP3 expression level promotes embryonic cancer cell apoptosis [34] and cardiomyocyte apoptosis during myocardial infarction [35]. These studies proposed that FABP3 overexpression reduces mitochondrial activity characterized by lower ATP synthesis and a lower mitochondrial membrane potential, as well as increased reactive oxygen species (ROS) levels and abnormal mitochondrial morphology. FABP3 is also critical for the loss of mitochondrial activity and ROS production during the neurodegenerative process in dopaminergic neurons [36]. In contrast, FABP5 expression is elevated in CA1 neurons (which are apoptotic after ischemia), but was unchanged in DG neurons that remained relatively stable after ischemia in the hippocampus of adult monkeys, suggesting that FABP5 is likely involved in the survival of hippocampal neurons [37]. 

Furthermore, FABP7 can participate in the proliferation of hippocampal astrocytes after ischemia [37]. However, it remains unclear whether the ischemia-induced enhancement of reactive astrocytes and FABP7 levels elicit beneficial or deleterious effects in neurons [38,39]. Astrocytes expressing FABP7 are crucial for the normal development of dendritic arbors, the formation of and transmission through the excitatory synapses of cortical neurons [40], and even increases the loss of ventral horn neurons in FABP7-knockout mice with spinal cord injury [41]. Conversely, FABP7 overexpression can directly promote a nuclear factor-kappa B (NF-κB)-driven pro-inflammatory response in astrocytes of mice with amyotrophic lateral sclerosis and can ultimately reduce motor neuron survival [42], whereas FABP7 knockdown in the developing brain can increase the proportion of neurons [43,44]. Based on these findings, we hypothesized that FABPs elicit detrimental effects on mitochondrial homeostasis and neuroinflammation, whereas FABP7 is partially required for neuronal differentiation in the developmental stages. However, further studies are required to identify the roles of FABP3, FABP5 and FABP7 after I/R injury.

In addition, we identified important roles for the elevated FABP levels in non-ischemic areas (left hemispheres) after brain ischemia. Previous reports indicated that ischemic lesions cause metabolite changes [45], edema and decreased cerebral blood flow [46] in remote non-ischemic regions. However, in non-ischemic areas, ischemia does not directly cause these changes, and some intermediate incentives are necessary to explain the induction processes. In addition to cerebral ischemia, many findings have also confirmed that in other neurodegenerative disorders (such as Alzheimer’s disease and Parkinson’s disease), cerebrospinal fluid (CSF) FABP3 levels [47,48] and serum FABP7 levels [49] are elevated in patients, and FABP5 is upregulated in a mouse model of Alzheimer’s disease [50]. Since neuroinflammation is a key contributing factor for neurodegenerative diseases, the high levels of pro-inflammatory cytokines, such as tumor necrosis factor-alpha (TNF-α), interleukin (IL)-1β and IL-6 [51,52,53], are found in the brain and CSF of patients. We speculated that pro-inflammatory cytokines spread from the ischemic core lesion to non-ischemic areas. FABP5 expression is upregulated by lipopolysaccharide (LPS) in human lung epithelial BEAS-2B cells [54] or by TNF-α in human aortic endothelial cells [55]. FABP7 levels are upregulated in spinal cord astrocytes in a mouse model of experimental autoimmune encephalomyelitis [56] and in primary cortical neuronal cultures after exposure to glutamate [57]. Taken together, FABPs trigger the neuronal injury in non-ischemic regions. However, further studies are required to confirm the FABP-induced neuronal injury.

In this study, we administrated MF6 to treat I/R injury in mice, and MF6 effectively reduced infarct volumes and lessened neurological dysfunction, suggesting that it can potentially be used as a therapeutic drug for ischemic injury. Regarding the site of action of MF6, we previously determined that MF6 significantly inhibited FABP3 and FABP5 protein expression and had a high affinity for both proteins [23]. Based on the above results, we envisioned the mechanisms underlying the ameliorative effects of MF6 on an ischemic injury. After administering MF6, it quickly entered the brain through the BBB and associated with FABP3 and FABP5 to different degrees to block their activities, which inhibited FABP-induced signaling pathways such as apoptosis and inflammation [8,34]. Consequently, the degree of cerebral ischemic injury was reduced, and then the weakened injury in turn lowered the stimulation of FABP expression levels. 

We recently found that FABP5 causes cell death under oxidative stress in glial cells [58]. When cells were exposed with psychosine, a phospholipase A2 activator, it caused the mitochondria-induced glial death via forming mitochondrial macropores with voltage-dependent anion channel (VDAC-1) and BAX [58]. The MF6 treatment completely abolished the psychosine-induced mitochondrial injury. Even though cell type expressed FABP5 are different between neuronal cells in brain ischemia in the present study and glial cells treated with psychosine by Cheng et al, FABP5-mediated mitochondrial injury in part mediates the neuronal death in the brain ischemia. 

FABPs themselves may participate in the FABP upregulation, because MF6 inhibited the FABP5 transcriptional activity induced by AA. AA is known to elevate in during ischemia [59] and it is also a strong ligand of FABP3 and FABP5 [23]. Moreover, AA is an endogenous activator of PPARs [60]. In this context, we speculate that FABPs transport AA from the cytoplasm to the nucleus to promote the activities of PPARs [61]. 

In addition, there are several the binding sites (PPRE) of PPARs on the promoter region of FABP3, FABP5 and FABP7 genes and the activations of PPARβ/δ and PPARγ mediate the expression of FABP5 [30,62]. Taken together, MF6 may reduce functions of FABPs, thereby inactivating transcriptional activities through PPARs. However, to resolve the precise interaction between PPAR subclasses (α, β/δ and γ) and FABP isoforms in neuronal cells, further extensive studies are required.

As a major inflammatory mediator, PGE_2_ is involved in the development of injury in several inflammatory neurological diseases, such as stroke and Alzheimer’s disease [63,64]. In this study, we confirmed that ischemia induced PGE_2_ accumulation in the brain, which was caused by the activation of mPGES-1, as previously reported [31]. Since COX-2 level is not changed in the non-ischemic regions [31], the increased mPGES-1 in the contralateral region may count for the increased PGE_2_ level. Since PPARγ activation enhances FABP5 mRNA expression and increase the levels of mPGES-1 mRNA and PGE_2_ in porcine trophoblast cells, and a PPARγ antagonist blocked their up-regulation [62], the regulation of the FABP5 and mPGES-1 upregulation may depend on PPARγ. Furthermore, FABP5 inhibition reduces the nuclear transport of NF-κB, thereby decreasing mPGES-1 expression and PGE_2_ synthesis [65], NF-κB also induces the expression of various pro-inflammatory genes [66] and induces neuronal death in cerebral ischemia [67]. In this context, both PPAR and NF-κB may underly the mechanism of upregulation of mPGES-1–PGE_2_ in brain ischemia. Moreover, our findings showed that administering MF6 inhibited FABP3 and FABP5 expression and strongly prevented the induction of mPGES-1 expression and PGE_2_ levels in the brain of I/R mice, indicating that the activation of FABP–mPGES-1–PGE_2_-signaling pathways promotes ischemic injury. However, it is unclear whether one FABP or FABP3, FABP5 and FABP7 collectively induced mPGES-1 after ischemia. This question requires further investigation, but our data indicated that at least FABP5 was present. Further extensive studies are required to define whether FABP3 or FABP7 is involved in PPAR and NF-κB signaling and whether MF6 treatment inhibits the activity of NF-κB.

Our present results first defined that FABP inhibitor MF6 can inhibit ischemic brain injury by inhibiting the expression and function of activated FABPs after ischemia. Further extensive studies are required to confirm the detrimental functions of FABPs in neuron and glia by using the knockout mice of FABPs. In addition, MF6 has high affinity for FABP7 as compared to FABP3 and FABP5. Since FABP7 is extensively expressed in the astrocytes, the effect of MF6 on the PGE_2_ production should be defined in the astrocytes after brain ischemic injury.

In summary, the present study suggested that increased FABP3, FABP5 and FABP7 expression in the brain is a novel biochemical marker of cerebral ischemia and that the FABP inhibitor, MF6 inhibited their expression levels to play a neuroprotective role in cerebral I/R in mice.

## Figures and Tables

**Figure 1 biomedicines-09-00529-f001:**
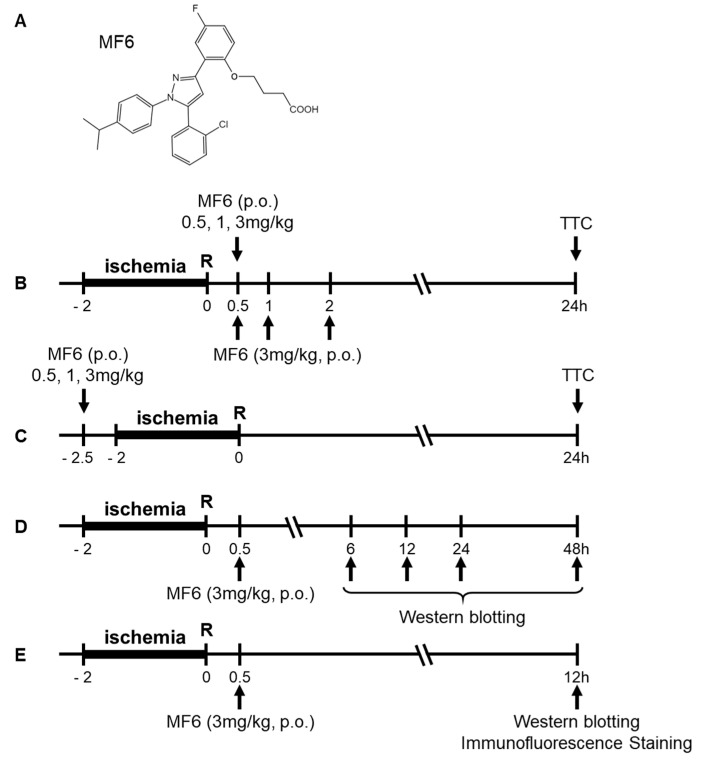
Experimental schedule for MF6 administration (0.5, 1 or 3 mg/kg, p.o.) in I/R mice. (**A**) Chemical structure of MF6. (**B**) ICR mice were subjected to tMCAO for 2 h. Subsequently, they were administered orally (p.o.) different concentrations of MF6 (0.5, 1 or 3 mg/kg) 30 min after reperfusion (for dose–response experiments) or 3 mg/kg MF6 at 0.5, 1 and 2 h after reperfusion (for time–response experiments). TTC staining was performed at 24 h after reperfusion. (**C**) MF6 (0.5, 1 or 3 mg/kg) was administered p.o. to mice 30 min before reperfusion to study the effects of pre-treatment. (**D**) MF6 (3 mg/kg) was administered to mice 30 min after reperfusion, and FABP levels were measured after 6, 12, 24 and 48 h. (**E**) I/R mice were administered MF6 (3 mg/kg) 30 min after reperfusion, and after 12 h, the brains were collected for Western blot and immunostaining analyses.

**Figure 2 biomedicines-09-00529-f002:**
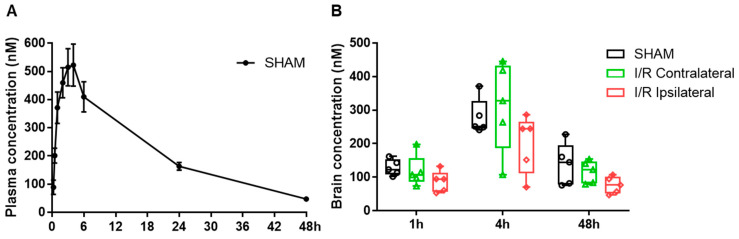
Pharmacokinetics of MF6 in the plasma and brain in sham and I/R mice. (**A**) MF6 plasma concentrations were measured at 0.25, 0.5, 1, 2, 3, 4, 6, 24 and 48 h after MF6 administration in sham mice (n = 5 per group). The data shown in each represent the mean ± SEM. (**B**) The MF6 concentrations in the contralateral and ipsilateral regions of the brains of sham and I/R mice were measured at 1, 4 and 48 h after MF6 administration (n = 5 per group). No statistical difference was observed among the groups at the same time point by one-way analysis of variance (ANOVA) followed by Dunnett’s test.

**Figure 3 biomedicines-09-00529-f003:**
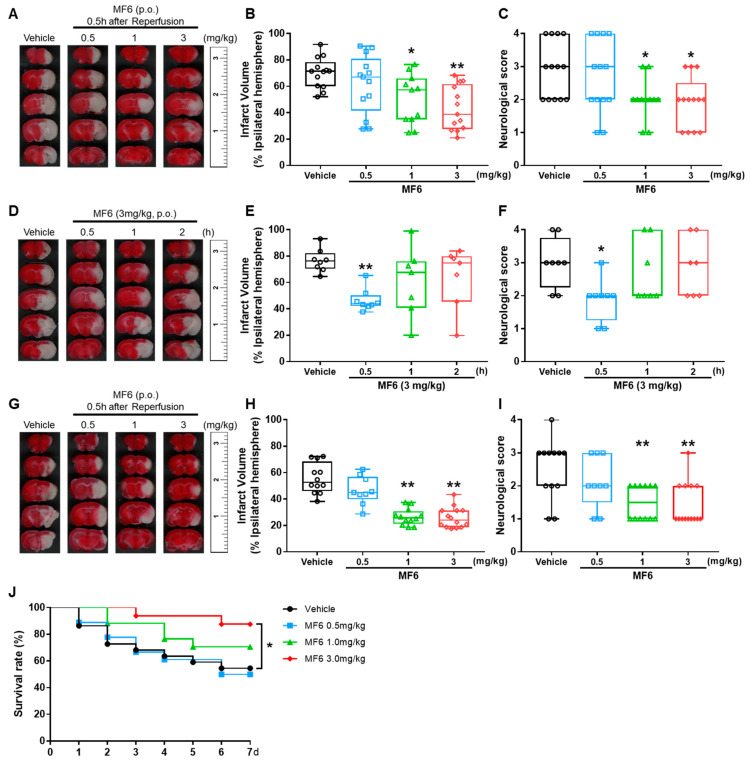
Effects of MF6 post-treatment on I/R injury in mice. Mice were subjected to tMCAO for 2 h. (**A**–**C**) MF6 was administrated at different concentrations (0.5, 1 or 3 mg/kg) 30 min after reperfusion. Representative images of TTC staining (**A**), quantitative analysis of the infarct volumes (**B**) and neurological deficits (**C**) at 24 h after reperfusion (n = 11–13). (**D**–**F**) MF6 (3 mg/kg) was administrated at 0.5, 1 or 2 h after reperfusion. Representative images of TTC staining (**D**), quantitative analysis of infarct volumes (**E**) and neurological deficits (**F**) at 24 h after reperfusion (n = 7–8). (**G**–**J**) MF6 was administrated with different concentrations (0.5, 1, 3 mg/kg) 30 min after reperfusion. Representative images of TTC staining (**G**), quantitative analysis of infarct volume (**H**) and neurological deficits (**I**) at day 7 after reperfusion. The number of living animals in each group every day was recorded, and survival rate (**J**) was calculated (n = 9–14). * *p* < 0.05, ** *p* < 0.01 vs. the I/R-treated group with the CMC group (vehicle). Differences were statistically analyzed using one-way analysis of variance (ANOVA) followed by Dunnett’s test.

**Figure 4 biomedicines-09-00529-f004:**
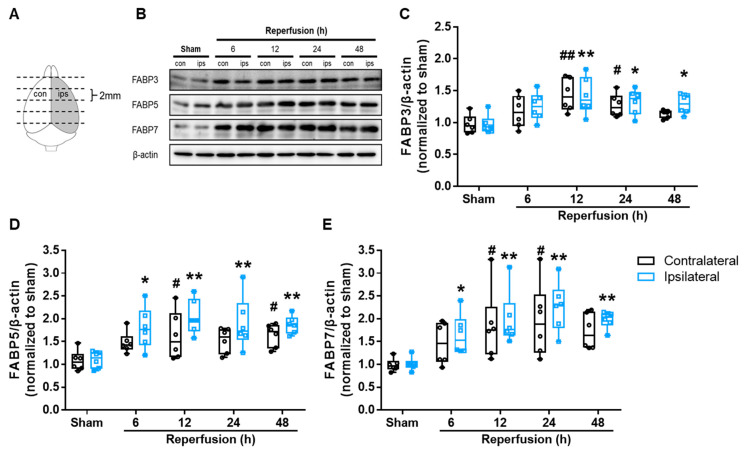
Effects of I/R injury on FABP3, FABP5 and FABP7 expression levels in the brain. (**A**) The locations of samples. Brains were cut into four slices (2-mm thick) from the front of the cortex. The second slices (designated as con and ips slices) of the brain, including the cortex and striatum, were used for Western blot analysis and PGE_2_-content analysis in the following experiments. (**B**–**E**) Mice were subjected to right tMCAO for 2 h. At 6, 12, 24 and 48 h after reperfusion, the second slice of the right brain (ipsilateral) and the left brain (contralateral) areas were collected for Western blot analysis of FABPs levels. (**B**) Representative images of Western blots. Quantitative analyses of FABP3 (**C**), FABP5 (**D**) and FABP7 (**E**) expression levels in contralateral (black) and ipsilateral (blue) areas of the brains. ^#^
*p* < 0.05, ^##^
*p* < 0.01 vs. the sham group (contralateral); * *p* < 0.05, ** *p* < 0.01 vs. the sham group (ipsilateral) (n = 6 per group). Differences were statistically analyzed using one-way analysis of variance (ANOVA) followed by Dunnett’s test.

**Figure 5 biomedicines-09-00529-f005:**
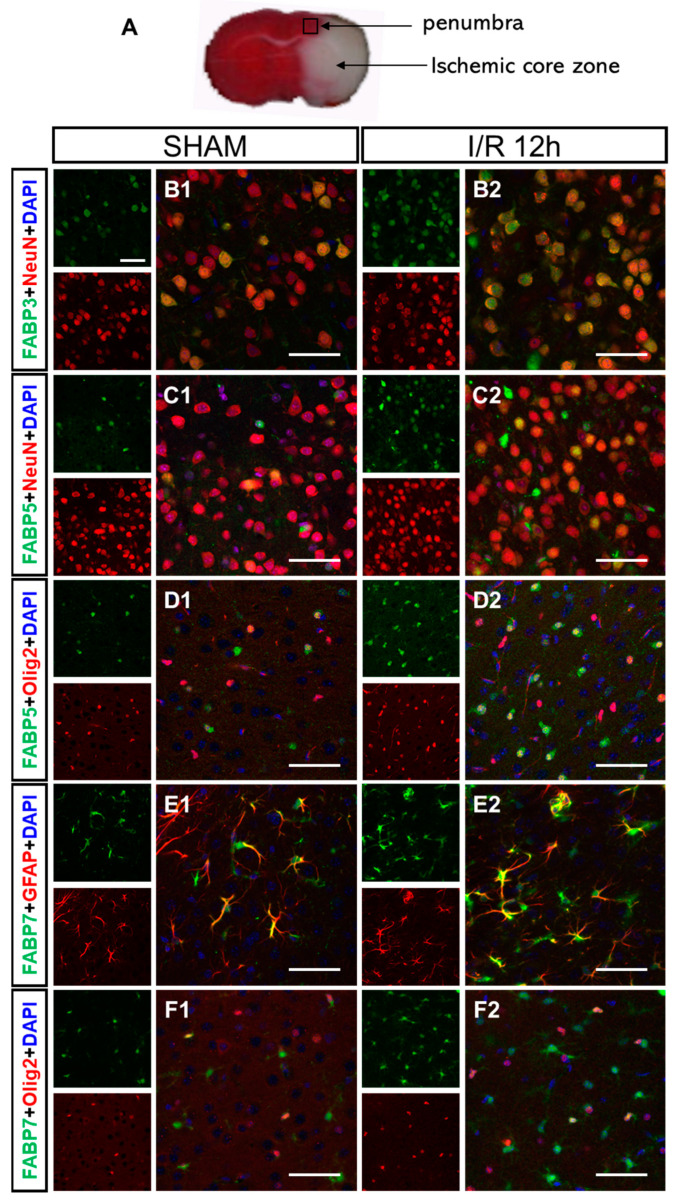
Immunofluorescence of FABP3, FABP5 and FABP7 in the cortexes of sham and I/R mice. (**A**) Representative micro-graphs of immunofluorescence staining of the cortical penumbra region (shown in the black box area) at 12 h after reperfusion. (**B**) Double staining for FABP3 (green) and NeuN (a neuronal marker, red) expression in sham mice (**B1**) and I/R mice (**B2**, ipsilateral). (**C**,**D**) Double staining for FABP5 (green) and NeuN (red; **C**) or Olig2 (an oligodendrocyte marker, red; **D**) in sham mice and I/R mice (ipsilateral). (**E**,**F**) Double staining for FABP7 (green) and GFAP (an as-trocyte marker, red; **E**) or Olig2 (red; **F**). Scale bar = 50 μm. The two small images on the left show immunofluorescence for FABPs and cell markers, whereas the larger image on the right is a merged image.

**Figure 6 biomedicines-09-00529-f006:**
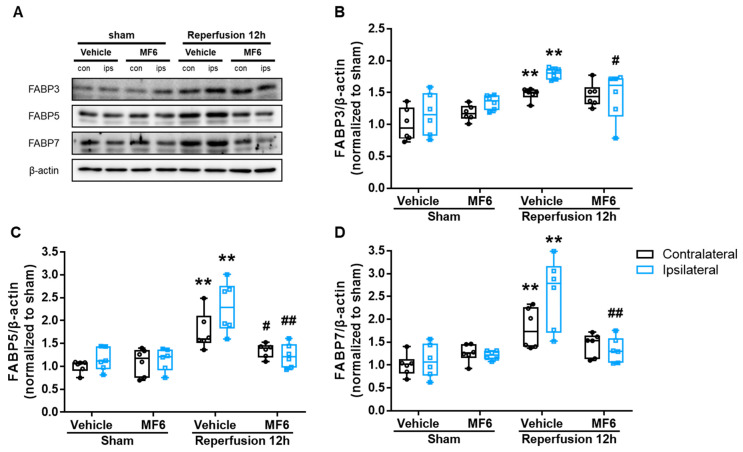
Effects of MF6 administration on FABP3, FABP5 and FABP7 expression levels in I/R mice. Mice were subjected to right tMCAO for 2 h and administered 3 mg/kg MF6 at 30 min after reperfusion. At 12 h after reperfusion, the second slice of the right (ipsilateral) and left (contralateral) brain areas, including the cortex and striatum, were used for Western blot analysis of FABP levels. (**A**) Representative images of Western blots. Quantitative analyses of FABP3 (**B**), FABP5 (**C**) and FABP7 (**D**) proteins expression levels in the contralateral (black) and ipsilateral (blue) regions of the brains. ** *p* < 0.01 vs. the sham-treated group, administered the vehicle (contralateral/ipsilateral); ^#^
*p* < 0.05, ^##^
*p* < 0.01 vs. the I/R-treated group, administered the vehicle (contralateral/ipsilateral) (n = 6 per group). Differences were statistically analyzed using two-way analysis of variance (ANOVA) followed by Student–Newman–Keuls test.

**Figure 7 biomedicines-09-00529-f007:**
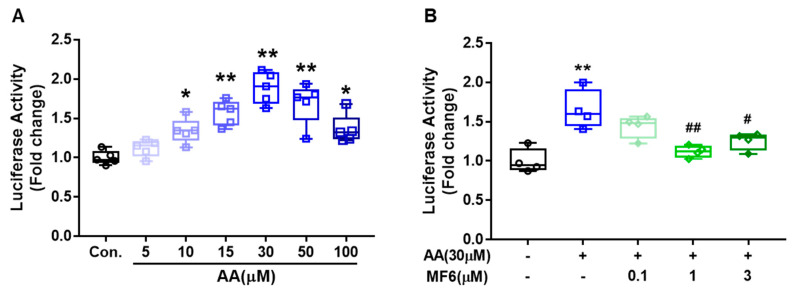
Effects of MF6 administration on AA-induced FABP5 transcriptional activity in SH-SY5Y cells. SH-SY5Y cells were co-transfected with FABP5-pGL3 and Renilla luciferase reporter vectors for 6h. (**A**) After transfection, cells were stimulated with different doses of AA (5 μM, 10 μM, 15 μM, 30 μM, 50 μM or 100 μM) for 24 h (n = 5 per group). (**B**) After transfection, cells were treated with AA/BSA and MF6 (0.1 μM, 1 μM or 3 μM) at the same time for 24 h (n = 4 per group). Promoter activity were represented as firefly/Renilla luciferase activity and were normalized to control cells (without AA treatment). * *p* < 0.05, ** *p* < 0.01 vs. the BSA-treated cells (Con.); ^#^
*p* < 0.05, ^##^
*p* < 0.01 vs. the AA and vehicle-treated cells. Differences were statistically analyzed using Student’s *t*-test or one-way analysis of variance (ANOVA) followed by Dunnett’s test.

**Figure 8 biomedicines-09-00529-f008:**
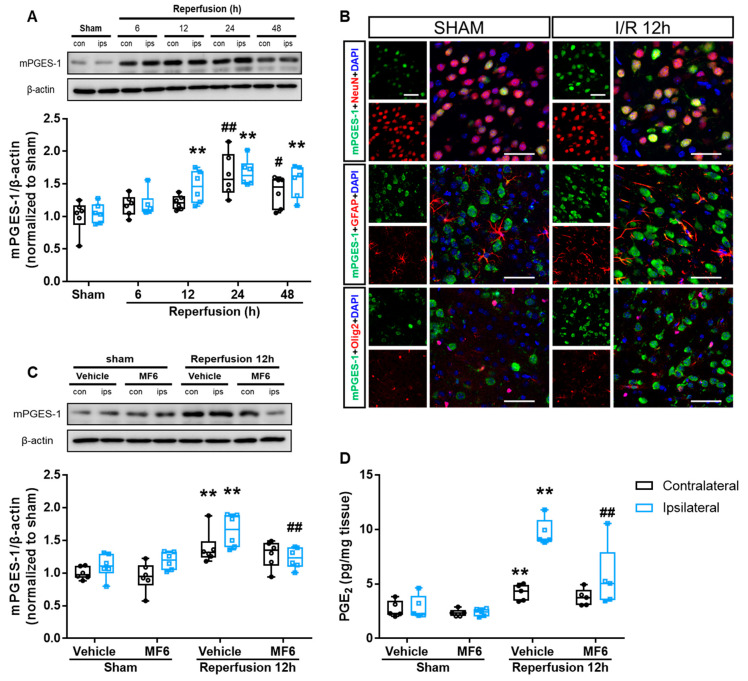
Effects of MF6 administration on mPGES-1 expression and PGE_2_ levels in I/R mice. (**A**) Mice were subjected to tMCAO for 2 h. Representative Western blot images and quantitative analyses of mPGES-1 protein expression in contralateral and ipsilateral brain regions at 6, 12, 24 and 48 h after reperfusion. ^#^
*p* < 0.05, ^##^
*p* < 0.01 vs. the sham group (contralateral); ** *p* < 0.01 vs. the sham group (ipsilateral) (n = 6 per group). (**B**) Representative micrographs of immunostaining for mPGES-1 (green) and NeuN, GFAP or Olig2 (red) in the cortical penumbra region at 12 h after reperfusion. Scale bar = 50 μm. (**C**) Mice were subjected to tMCAO for 2 h and administered 3 mg/kg MF6 at 30 min after reperfusion. Representative Western blot images and quantitative analyses of mPGES-1 protein expression in contralateral and ipsilateral brain regions at 12 h after reperfusion in I/R mice, treated with MF6 30 min after reperfusion (n = 6 per group). (**D**) PGE_2_ levels in contralateral and ipsilateral regions (Figure 4A) of sham and I/R mice treated with or without MF6 (n = 5 per group). ** *p* < 0.01 vs. the sham-treated group, administered the vehicle (contralateral/ipsilateral); ^#^
*p* < 0.05, ^##^
*p* < 0.01 vs. the I/R-treated group, administered the vehicle (contralateral/ipsilateral). Differences were statistically analyzed using one-way analysis of variance (ANOVA) followed by Dunnett’s test (**A**) or two-way analysis of variance (ANOVA) followed by Student–Newman–Keuls test (**C**,**D**).

## Data Availability

The data presented in this study are available on request from the corresponding author.

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
