# Peer review of "Fatty Acid-Binding Proteins Aggravate Cerebral Ischemia-Reperfusion Injury in Mice"

_biomedicines, 2021, doi:10.3390/biomedicines9050529_

Round 1
Reviewer 1 Report
Having read the manuscript entitled “Fatty acid-binding proteins aggravate cerebral ischemia-reperfusion injury in mice” I have to admit that the topic is interesting and the manuscript is well-written. However, before publication I have some issues that should be explained/clarified:
Introduction
In the Introduction sections the Authors should focus more on their study, i.e. the hypotheses of the study, performed tests, meaning of the study.
2.1. Animals
How many animals were kept in 1 cage? How many animals were used in total?
2.3. Drug treatment
On what basis did the Authors select the tested doses of MF6 and the time intervals (i.e., 30 min) between reperfusion and oral administration?
Results
In the Result section only the obtained results should be presented. Therefore information related to the Material and methods section or conclusions should be either removed or transferred to the appropriate sections.
Discussion
In the Discussion section a separate paragraph on the limitations of the study should be added.
Figure captions
The tested dose of MF6 and its route of administration should be added into the Figure 1 caption. Name of the statistical test should be added into figure captions.
References
References as old as from 1989, 1997, 1989, 2001, 2002, 2003, 2004 should be replaced by more recent ones.
There are several editorial errors that should be corrected.
Author Response
Response to Reviewer 1
- Introduction: In the Introduction sections the Authors should focus more on their study, i.e. the hypotheses of the study, performed tests, meaning of the study.
Ans: According to the comment, we described about our previous studies and meaning of the studies (Page 2, lines 12-33.
- 1. Animals: How many animals were kept in 1 cage? How many animals were used in total?]
Ans: Seven mice were kept in each cage. In the whole experiments, we used approximately 300 mice. We have been working to minimize the number of mice as feasible while retaining sufficient cohort sizes.
- 3. Drug treatment: On what basis did the Authors select the tested doses of MF6 and the time intervals (i.e., 30 min) between reperfusion and oral administration?
Ans: Previous studies in our laboratory have demonstrated that FABP3 ligands MF1 and MF8 at a concentration of 1.0 mg/kg showed statistically significant improvement effects in a mouse model of Parkinson’s disease [19, 20]. Since MF6 has similar Kd value for FABP3, we selected the doses of MF6 at 0.5 mg/kg, 1.0 mg/kg and 3.0mg/kg in this study. Our results showed that MF6 at 3.0 mg/kg had appropriate therapeutic effects. Therefore, we chosen the dose of 3.0 mg/kg as the optimum concentration for the subsequent experiments.
As for the timing of MF6 administration after reperfusion, we decided the timing to compare with our previous study. Protein tyrosine phosphatase 1B inhibitor KY226 only significantly reduces the infarct volume at 30 minutes after reperfusion, but not at 1 h or 2 h [24]. Further extensive studies are required to determine the therapeutic time window in MF6 therapeutics.
- Results: In the Result section only the obtained results should be presented. Therefore information related to the Material and methods section or conclusions should be either removed or transferred to the appropriate sections.]
Ans: According to the comment, we removed the descriptions related to the "Materials and Methods" section or conclusions.
- Discussion: In the Discussion section a separate paragraph on the limitations of the study should be added.]
Ans: According to the comment, we described the limitation of the present study and future direction to eolve the questions (page 16, line 1-7)
- Figure captions: The tested dose of MF6 and its route of administration should be added into the Figure 1 caption. Name of the statistical test should be added into figure captions.]
Ans: According to the comment, we added the dose and route and described the statistical test procedures in figure legends.
- References: References as old as from 1989, 1997, 1989, 2001, 2002, 2003, 2004 should be replaced by more recent ones.
Ans: According to the comment, we replaced the old references with new articles as listed below.
We replaced "Longa, E. Z. et al. 1989" with "Sun, M. et al. 2018" [23].
We replaced "Clark, W. M. et al. 1997" with "Sun, M. et al. 2018" [23].
We replaced "Liu, Y. et al. 2000" with "Matsumata, M. et al. 2012" [31].
We replaced "Allan, S. M. et al. 2001" with "Wang, W. Y. et al. 2015" [51].
We replaced "Gold, L. et al. 2002" with "Xu, Z. N. S. et al. 2013" [45].
We replaced "Izumi, Y. et al. 2002" with "Xu, Z. N. S. et al. 2013" [45].
We replaced "Lo, E. H. et al. 2003" with "Terasaki, Y. et al. 2014" [3].
We replaced "Basso, M. et al. 2004" with "Mollenhauer, B. et al. 2007" [13].
- There are several editorial errors that should be corrected.
Ans: According to the comment, the English of manuscript was edited by native speaker in Editage Inc.
Reviewer 2 Report
The authors of the manuscript “Fatty acid-binding proteins aggravate cerebral ischemia-reperfusion injury in mice” provided new interesting data for the role of FABPs in the brain under ischemic conditions. The authors claim that inhibition of these proteins by the chemical agent MF6 improves reperfusion-induced injury in mice subjected to transient middle cerebral artery occlusion. Overall the study is well conceived, and the experiments well designed, but few issues should be addressed to make this study more convincing and solid.
Major:
- In figure 5, the authors showed micrographs including immunofluorescence data of FABP3, FABP5, and FABP7 in the ipsilateral cortexes of sham and I/R mice. They included markers of neurons, oligodendrocytes, and astrocytes. In the supplementary data, other panels including co-staining with Iba1+ and FABPs microglia are provided. It’s not clear why the authors did not present a complete IF co-staining of all glial and neuronal markers with each FABP, e.g. FABP3 + Olig2+ GFAP, or FABP5 + GFAP+ NeuN. Such work together with cell quantification (percentage of each cell type positive for FABP) would provide a compelling evidence for the differential involvement of each glial or neuronal cell in I/R. In addition, it would very interesting to extend such data to the contralateral hemisphere as the authors already showed in figure 4 that FABPs can increase in the contralateral not infarcted cortex.
- The authors should discuss the increase of FABPs in non-infarcted regions of the brain more extensively in the discussion, and provide a new explanation for this phenomenon for I/R
- The authors reported an increase of mPGES-1b in NeuN+ neurons in mice undergone to tMCAO. In figure 8D, a staining including both neuronal and glial markers of both contralateral and ipsilateral hemisphere should be added.
- In the same figure, mPGES-1b protein increases in both contralateral and ipsilateral hemisphere. This phenotype seems to be in contradiction with the PGE2 levels represented in the same figure. Please provide a compelling explanation in the discussion
Minor:
- In the manuscript the authors typed a dash between syllables of some words. An example is in page 2, line 3, 8, and 9. Please check carefully the entire manuscript for such errors.
- In M&M the authors should specify what ICR (mice) stands for
- Please add a scale bar in the representative images of TTC staining
- Page 15, line 5. I bet the authors meant phospholipase A2 instead of phoapholipase A2
Author Response
Response to Reviewer 2
Major:
- In figure 5, the authors showed micrographs including immunofluorescence data of FABP3, FABP5, and FABP7 in the ipsilateral cortexes of sham and I/R mice. They included markers of neurons, oligodendrocytes, and astrocytes. In the supplementary data, other panels including co-staining with Iba1+ and FABPs microglia are provided. It’s not clear why the authors did not present a complete IF co-staining of all glial and neuronal markers with each FABP, e.g. FABP3 + Olig2 + GFAP, or FABP5 + GFAP + NeuN. Such work together with cell quantification (percentage of each cell type positive for FABP) would provide a compelling evidence for the differential involvement of each glial or neuronal cell in I/R. In addition, it would very interesting to extend such data to the contralateral hemisphere as the authors already showed in figure 4 that FABPs can increase in the contralateral not infarcted cortex.
Ans: Thank you for useful suggestion that each FABP should co-staining with all glial and neuronal markers to measure the percentage of each cell type positive for FABP. We would like to solve the problem regarding the source of antibodies. We used specific antibodies FABP3 (host species, mouse) + Olig2 (mouse) + GFAP (rabbit) and are not suitable for co-staining. We will resolve the problems future studies and provide the data of cell quantification (percentage of each cell type positive for FABP). Likewise, the changes of FABPs in the contralateral hemisphere are very important to verify the protective mechanism of MF6. We will answer the question in the next step.
- The authors should discuss the increase of FABPs in non-infarcted regions of the brain more extensively in the discussion, and provide a new explanation for this phenomenon for I/R.
Ans: According to the suggestion, we discussed the increased FABPs in non-infarcted regions as followed. “Since neuroinflammation is a key contributing factor for neurodegenerative diseases, the high levels of pro-inflammatory cytokines, such as tumor necrosis factor-alpha (TNF-α), interleukin (IL)-1β and IL-6 [51-53], are found in the brain and CSF of patients. We speculated that pro-inflammatory cytokines spread from the ischemic core lesion to non-ischemic areas. FABP5 expression is upregulated by lipopolysaccharide (LPS) in human lung epithelial BEAS-2B cells [54] or by TNF-α in human aortic endothelial cells [55]. FABP7 levels are upregulated in spinal cord astrocytes in a mouse model of experimental autoimmune encephalomyelitis [56] and in primary cortical neuronal cultures after exposure to glutamate [57]. Taken together, FABPs trigger the neuronal injury in non-ischemic regions. However, further studies are required to elicit the mechanism underlying FABP-induced neuronal injury.”(Page14 line 47 to page 15, line2)
- The authors reported an increase of mPGES-1b in NeuN+ neurons in mice undergone to tMCAO. In figure 8D, a staining including both neuronal and glial markers of both contralateral and ipsilateral hemisphere should be added.]
Ans: According to the comment, we added the data IF co-staining results (mPGES-1 + GFAP + DAPI, and mPGES-1 + Olig2 + DAPI) in Figure 8B.
- In the same figure, mPGES-1b protein increases in both contralateral and ipsilateral hemisphere. This phenotype seems to be in contradiction with the PGE2 levels represented in the same figure. Please provide a compelling explanation in the discussion.]
Response: We have added the following explanation of this phenotype in the discussion."Since COX-2 level is not changed in the non-ischemic regions [31], the increased mPGES-1 in the contralateral region may count for the increased PGE2 level." (page 15, line 36 to 38)
Minor:
- In the manuscript the authors typed a dash between syllables of some words. An example is in page 2, line 3, 8, and 9. Please check carefully the entire manuscript for such errors.
Ans: We have corrected such errors throughout the manuscript.
- In M&M the authors should specify what ICR (mice) stands for.
Ans: We added "(Institute of Cancer Research (ICR) mice, Slc:ICR)" in M&M. (page 2, line 36). If we understand incorrectly, please correct us again.
- Please add a scale bar in the representative images of TTC staining
Ans: According to the reviewer’s suggestion, we have added scale bar for the representative images of TTC staining in Figure 3 and Figure. S1.
- Page 15, line 5. I bet the authors meant phospholipase A2 instead of phoapholipase A2
Ans: "phoapholipase A2" was corrected to "phospholipase A2". (page 15, line 15)
Round 2
Reviewer 2 Report
The authors claimed that the co-staining FABP3 (host species, mouse) + Olig2 (mouse) + GFAP (rabbit) does not bear satisfying results. This is probably caused by the choice of using two antibodies sharing the same host species. Despite this poor convincing explanation, I trust the authors' decision to address this issue in the future and feel satisfied with the other comments to my raised issued